# Ecological Security Pattern Construction in Karst Area Based on Ant Algorithm

**DOI:** 10.3390/ijerph18136863

**Published:** 2021-06-26

**Authors:** Xiaoqing Zhao, Qifa Yue, Jianchao Pei, Junwei Pu, Pei Huang, Qian Wang

**Affiliations:** 1School of Earth Sciences, Yunnan University, Kunming 650500, China; yueqf123@126.com (Q.Y.); wangqian@mail.ynu.edu.cn (Q.W.); 2School of Mathematics and Statistics, Yunnan University, Kunming 650500, China; a1415300207@163.com; 3Institute of International River and Ecological Security, Yunnan University, Kunming 650500, China; pujunwei666@foxmail.com (J.P.); hphyyy09@126.com (P.H.)

**Keywords:** Karst area, ecological security pattern, Ant algorithm, the paradigm of “Source-Corridor-Ecological restoration point”, ecosystem services

## Abstract

Constructing the ecological security pattern is imperative to stabilize ecosystem services and sustainable development coordination of the social economy and ecology. This paper focuses on the Karst region in southeastern Yunnan, which is ecologically fragile. This paper selects the main types of ecosystem services and identifies the ecological source using hot spot analysis for Guangnan County. An inclusive consideration of the regional ecologic conditions and the rocky desertification formation mechanism was made. The resistance factor index system was developed to generate the basic resistance surface modified by the ecological sensitivity index. The Ant algorithm and Kernel density analysis were used to determine ecological corridor range and ecological restoration points that constructed the ecological security pattern of Guangnan County. The results demonstrated that, firstly, there were twenty-three sources in Guangnan County, with a total area of 1292.77 km^2^, accounting for 16.74% of the total. The forests were the chief ecological sources distributed in the non-Karst area, where Bamei Town, Yangliujing Township and Nasa Town had the highest distribution. Secondly, the revised resistance value is similar to “Zhe (Zhetu Township)-Lian (Liancheng Town)-Yang (Yangliujing Township)-Ban (Bambang Township)”. The values were lower in the north and higher in the south, which is consistent with the regional distribution of Karst. Thirdly, the constructed ecological security pattern of the “Source-Corridor-Ecological restoration point” paradigm had twenty-three ecological corridors. The chief ecological and potential corridor areas were 804.95 km^2^ and 621.2 km^2^, respectively. There are thirty-eight ecological restoration points mainly distributed in the principal ecological corridors and play a vital role in maintaining the corridor connectivity between sources. The results provide guidance and theoretical basis for the ecological security patterns construction in Karst areas, regional ecologic security protection and sustainable development promotion.

## 1. Introduction

In the 21st century, with the continuous advancement of urbanization, issues such as over-exploitation and inefficient land resources utilization, human-land relationship deterioration and spatial land-use pattern confusion [1,2,3,4,5] are becoming graver in karst area, which seriously threatens the coordinated development of regional ecological environment and social economy under the background of economic globalization [6]. Ensuring regional ecologic security to achieve sustainable development is a common problem faced by karst regions worldwide [7]. In this context, Ecological Security Pattern has gradually become an effective way to solve the problem. As per the mutual feedback of process and pattern principle, Ecological Security Pattern (ESP) can regulate and control process, maintain the structural integrity of the ecosystem, simulate the spatial movement in process, identify key landscape components, obtain multi-category, allow multi-level and multi-objective spatial allocation scheme [8,9]. The practical construction of ecological security patterns can gradually control the disorderly spread of cities and play a vital role in coordinating the orderly social economy development, ecologic environment development and ensuring regional ecologic security [10].

Early international research on ecological security patterns primarily focused on the protection of biodiversity [11], and then gradually began to study the value of ecosystem services [12]. The increasingly closes relationship between economic development and ecological quality propels society to be more attentive towards ecological protection, and the study of ecological security patterns gradually evolves into the coupling analysis of the social economy and natural ecology [13,14]. Currently, the pattern of “Source-Corridor” is formed. Among them, the ecological source area refers to the habitat patches which are of great significance to the regional ecological security or have radiation function, and it is the basis of constructing the ecological security pattern [15]. The identification of sources is through the selection of the nature reserves [16], reservoirs with stable land-use types and radiation functions, natural forest land [17] or constructing evaluation index systems such as habitat importance and landscape connectivity [18]. The traditional methods for source identification are easy and convenient, but it only considers the importance of ecological functions, functional degradation risk and neglect the internal differences of similar land type. Therefore, this study identified the important areas of ecosystem services by considering the quantitative evaluation of the supply capacity of regional typical ecosystem services, and then selected the key patches with important ecological service supply value as the ecological source to maintain the health and integrity of the regional landscape pattern and meet the ecological needs of human beings.

As an important landscape element, the ecological corridor retains the natural vegetation in degraded landscape [19,20], connecting discrete ecological source areas, promoting the effective flow of species between source areas and protecting regional biodiversity [21,22]. In addition, the edge of the ecological corridor can maintain the stability and integrity of the ecological process and reduce the interference of human activities on the regional ecosystem [23,24]. Therefore, the ecological corridor is not a line, but a belt with a specific spatial scope [25]. However, due to the landscape heterogeneity, the spatial scope of ecological corridor is still difficult to determine, which has gradually become a hot and challenging point in the extraction of the ecological corridor [26]. At present, the main corridor extraction models are the minimum cumulative resistance (MCR) model and the circuit theory model [27]. The principle of MCR model is that the optimal path of species transfer between source points is regarded as the path with the least cumulative resistance [28]. At the same time, the circuit theory transforms the ecological resistance into electric current and determines multiple ecological corridors [29,30] by simulating the random walk of species. Among them, MCR model can only extract the spatial location of ecological corridor but cannot calculate the width threshold of the ecological corridor. Circuit theory can only judge the corridor range subjectively, especially considering the influence of local background on species movement and can only set the corridor width to a specific range according to experience. Few scholars applied ant colony algorithm to determine the corridor range. Ant colony algorithm can simulate the process of ants searching for the best path between nest and nearby food resources [31,32]. In other words, when ants are looking for the optimal path, they will leave pheromones along the way to meet the needs of constructing continuous paths [26,33]. When pheromones diffuse in space, it provides quantitative support for identifying the scope of information flow in space. Through the guidance of pheromone, a positive feedback mechanism is formed between ants, which improves the accuracy and objectivity of calculation [34]. In recent years, the application of ant colony algorithm has gradually penetrated into various fields, including robot path planning [35], logistics distribution path selection [36], unmanned Lane planning [37], irrigation scheduling optimization [38]. It can be seen that ant colony algorithm is one of the means to solve the path optimization. Therefore, the application of ant colony algorithm can identify the width of ecological corridor objectively and accurately, reduce the subjective experience judgment of ecological corridor and make the results more in line with the regional reality. At the same time, the algorithm can flexibly change the corresponding parameters according to the characteristics of different research areas, which has strong generalization and practicability. In addition, the determination of the scope of the ecological corridor can highlight the discontinuous part of the ecological corridor in space, which is regarded as the ecological restoration point, which is of great significance to improve the connectivity of the ecological corridor and maintain the complete ecosystem.

From a global perspective, the influence of ancient hard carbonate rocks, water, heat matching of monsoon climate and unreasonable human interference in the Southwestern Karst region of China makes it extremely sensitive to ecology, reducing the ecological carrying capacity of the southwestern Karst region [39]. Furthermore, the insufficiency of congenital soil formation conditions for carbonate rocks and the shallow soil layer limits the vegetation growth in Karst areas and causes frequent rocky desertification and soil erosion [40]. Facing the dual pressures of protecting ecology and economic development in Karst areas and constructing an ecological security pattern is an effective way to stabilize the regional ecosystem service structure and prevent further landscape fragmentation. Therefore, this paper evaluates the Karst ecological environment status and the natural base characteristics-based chief ecosystem services and extracts the vital ecological patches as the ecological source using the hot spot analysis method. The construction of the basic resistance surface was as per the specific geographical location of the region and the rocky desertification formation mechanism, and the ecological sensitivity index modifies the fundamental resistance value, combined with the MCR model, Ant algorithm and Kernel density function to determine the scope of ecological corridors and ecological restoration points. This study aims to determine the Karst areas for focusing on the protection of the ecological sources. Further, it develops a comprehensive resistance surface based on the specific natural basement in the Karst area. Furthermore, this study corrects the Ant algorithm, determines the corridor range and ecological restoration points, and constructs the ecological security pattern. The results can maintain the biodiversity of the karst area, maintain the stability of the regional ecosystem, reduce the damage of unscientific human disturbance to the natural system to a certain extent and provide a reference for karst rocky desertification control and restoration of mountains-rivers-forests-farmlands-lakes-grasslands in Karst areas.

## 2. Data and Method

### 2.1. Study Area

Guangnan County is in the southeast of Yunnan Province, from 104°31′ to 105°39′ and 23°29′ to 24°28′ N, the border of Yunnan, Guizhou and Guangxi Provinces, and is a part of the Karst Plateau in the southeast of Yunnan (as shown in Figure 1). The land area is 7730.09 km^2^, with mountains, semi-mountains and hills widely distributed and covering 94.7% of the total county land area. The terrain is high in the southwest and low in the northeast, with an average elevation of 1280 m. The widely distributed Karst landscape covers about three-fourths of the county and is prone to soil erosion, Karst collapse and other natural disasters. Severe human disturbance superimposes fragile ecological environment, making the rocky desertification problem prominent and difficult to control, is one of the 200 chief rocky desertification counties in China.

Guangnan County is in the subtropical plateau monsoon climate zone, with hot rain in the same period and distinct dry season and rainy season. The Karst area has a wide distribution of carbonate rock, mainly composed of the transition layer of limestone and dolomite, with poor soil-forming conditions. The non-Karst area, a national natural forest protection county, is rich in vegetation and species diversity. Guangnan County is rich in water resources, including Xiyang River, Tuoniangjiang River, Qingshui River and Nanli River. The numerous difficulties in the conservation, exploitation and utilization of water resources make it challenging to fulfill residents’ daily life and production activities requirements.

### 2.2. Data Sources and Processing

The land-use status map compilation is as per the Landsat 8 Oli remote sensing image data in 2018 from the Geospatial Data Cloud (http://www.gscloud.cn./) with 15 m spatial resolution. ENVI5.3 (Harris Geospatial Solutions, Inc., Broomfield, CO, USA) processes the image with band combination, radiation calibration, atmospheric correction and others to eliminate image errors and combined with Google image map, the second national land use survey map and other auxiliary data, ArcGIS10.2 (ESRI, Redlands, CA, USA) using human-computer interaction interprets the image according to Chinese Land Use Status Classification Standard (GB/T21010–2017). The land-use is of eight types, i.e., forest land, grassland, farmland, garden, water area, unused land, transportation land and construction land. Furthermore, according to the lithological characteristics, the non-Karst area and the Karst area are two study area divisions, and rocky desertification degree in the Karst area was distinct, including severe rocky desertification, moderate rocky desertification, mild rocky desertification, potential rocky desertification and non-rocky desertification. Finally, selected 246 sample points were for field results verification, and the accuracy of land use type and rocky desertification degree was 90.45% and 85.11%, respectively, which is as per the research requirements.

Furthermore, DEM data comes from the Geospatial Data Cloud (http://www.gscloud.cn/), which extracts slope and elevate data. Meteorological data comes from Guangnan County Meteorological Bureau and China Meteorological Data Network (http://data.cma.cn/), obtained by spatial interpolation based on the rainfall data of the study area and surrounding meteorological stations, and calculate the water resources supply and soil conservation. Potential evapotranspiration comes from the National Ecosystem Observation and Research Network Science and Technology Resources Service System (http://www.cnern.org.cn.), used for water resources supply calculations. For soil conservation calculations, the soil data comes from the China Soil Data Set (v1.1) of the World Soil Database (HWSD) (http://westdc.westgis.ac.cn/) with a resolution of 1 km.

Food production and other statistical data are from the Guangnan County Statistical Yearbook (2000–2017), used for food supply calculations. The Karst type collapse-prone area comes from Guangnan County Natural Resources Bureau, used in the ecological sensitivity index calculation. Uniformly converted all spatial data is for the WGS_1984_UTM_48N coordinate system.

## 3. Research Methods

The ecological security patterns construction mainly has three stages. In the first step, after extracting the top 20% of the ecosystem service value, the ecological source extraction is by the hot spot analysis method. In the second step, the constructed index system determines the basic resistance surface and obtains the comprehensive resistance surface by the ecological sensitivity index modification. In the final step, an all-inclusive resistance surface, combined with the MCR model, Ant algorithm and Kernel Density analysis, determines the scope of ecological corridors and ecological restoration points and constructs the ecological security pattern. Figure 2 shows the specific research steps.

### 3.1. Identification of Ecological Sources

The sources refer to the patches that provide the chief ecosystem services in the region, which produce higher value for the landscape process development [39] and have ecological significance for maintaining biodiversity. Furthermore, the sources refer to the source point of species diffusion and maintenance, which plays a crucial role in maintaining the health and integrity of landscape patterns and meeting human ecological needs [40].

It is an effective way to determine the ecological source by quantitatively assessing the supply capacity of general regional ecosystem services, analyzing the value and differentiation of different ecological functions, identifying the vital areas of ecosystem services and then screening out the crucial patches of significant ecological service supply value [41]. Forest land and cultivated land are the main land cover types in Guangnan County, with sixteen large and small rivers belonging to the Pearl River and Red River systems with abundant water resources and diverse biological species. The quantitative evaluation of five ecosystem services of gas regulation, habitat maintenance, soil conservation, water resources supply and food supply was based on the study area’s ecological environment status and natural base characteristics (Table 1). The first selected 20% of each ecosystem service value is the chief ecological patch, and the use of the hot spot analysis method extracts the ecological source.

#### 3.1.1. Ecosystem Services

Specifically, gas regulation, one of the critical regulation services of terrestrial ecosystems, plays a vital role in tracking global warming and regulating changes and measurement is by the CASA model [42].

Habitat maintenance uses the biodiversity module of the InVEST model to judge the level of biodiversity through the habitat maintenance level [43].

Soil conservation plays a crucial role in maintaining ecological security and controlling regional soil erosion, calculated by the revised general equation of soil and water loss [44].

Water resources supply estimation is by the water production module of the InVEST model, based on the water balance method [45].

Obtaining food supply is by spatial interpolating the grain output in the statistical yearbook according to the normalized vegetation index (NDVI) value of the cultivated land in each county of Guangnan County.

#### 3.1.2. Hot Spot Analysis

The ecological patch area determined by the ecosystem services is small, the degree of connectivity is low and the radiation function is weak. The Getis-Ord Gi * statistical method [46] based hot spot analysis method can conduct the spatial cluster analysis of high and low values for the ecosystem services. Therefore, the hot spot analysis eliminates the patches with higher fragmentation and lower concentration, and the high concentration, close distance and large-scale patches are the ecological sources.

### 3.2. Determination of Landscape Resistance Surface

#### 3.2.1. Determination of Resistance Factors and Basic Resistance Coefficient

Ecological resistance is an obstacle in the energy transmission process, material exchange or species migration between sources. The resistance surface construction is the core of the ecological security pattern, and land use type and topography are the chief sources of resistance to the “source” spread out [47]. The study area has hills, mountains and semi-mountains with large undulations and high-altitude differences. The topography and types of land cover can control the soil and hydrological conditions that determine the spatial landscape differentiation in the area. The study area is a general Karst area in the southwest plateau of China. Due to the old and hard carbonatite and the congenitally deficient soil-forming conditions, the local soil layer becomes shallow and thin, the continuity of soil cover is meager, the nutrient is scarce and easy to run off, and water seepage is fast, which limits the Karst mountain vegetation productivity. Moreover, with the influence of subtropical plateau monsoon climate and human interference, the vegetation coverage becomes low with the majority of secondary dwarf forests and shrubs and rocky desertification in some areas [48]. The entropy weight method determines the weight of the six selected resistance factors, i.e., land cover type, slope, altitude, vegetation coverage, soil thickness and bedrock type. As per the research [49,50,51,52,53,54], determining the basic resistance coefficient for each factor, the resistance coefficient size reflects the difficulty degree of ecological source diffusion, and Table 2 shows the specific basic resistance coefficient.

#### 3.2.2. Correction of the Basic Resistance Surfaces

As the resistance value relates to regional ecologic risk, fragile ecosystems and frequent natural disasters have split the contiguous ecological sources and destroyed the original ecological corridors. The study area has a shallow soil layer and abundant and uneven precipitation. Over 60% of the rain occurs between June and August, which results in frequent soil erosion and Karst collapse, and rapidly deteriorating rocky desertification, leading to domestic species liquidity and spatial liquidity decline. Therefore, the ecological resistance value determination comprehensively considers the impact of rocky desertification degree on Karst collapse and soil erosion. An ecological sensitivity index construction corrects the basic resistance surface, and the calculation method is as follows:(1)$$R_{i}=R\times \frac{ESI}{ESI_{mean}}$$
(2)$$ESI=a\times RDI_{i}+KCI_{i}+c\times SEI_{i}$$
(3)$$ESI_{\mathrm{mean}}=\frac{ESI-ESI_{\min}}{ESI_{\max}-ESI_{\min}}$$Here *R_i_* is the corrected resistance coefficient, *R* is the basic resistance coefficient, *ESI_mean_* is the Ecological Sensitivity Index after standardized treatment and *ESI* is the Ecological Sensitivity Index. The *RDI_i_*, *KCI_i_* and *SEI_i_* are rocky desertification index, Karst collapse index and soil erosion index, respectively, with a, b and c weights for each fac0.483, 0.282 and 0.235, respectively calculated by entropy weight method *ESI*_min_ and *ESI*_ma*x*_ are the minimum and maximum of ecological sensitivity index, respectively.

As per the corrected resistance coefficients, the minimum cumulative resistance surface uses the *MCR* model [55]. In the *MCR* model, the resistance value does not represent the distance between two points, but the cumulative cost distance across two points is as follows:(4)$$MCR=f\min\sum_{j=n}^{i=m}D_{ij}\times R_{i}$$Here *MCR* is the Minimum Cumulative Resistance value, *f* is the positive correlation between the ecological process and the *MCR*, *D_ij_* is the spatial distance from the ecological source patch *j* to the landscape unit *I* and *R_i_* refers to the corrected resistance coefficient.

#### 3.2.3. Scope of Ecological Corridors and Ecological Restoration Points

A channel between ecological sources, categorizing and protection of ecological corridors are imperative for promoting regional ecologic elements circulation and biodiversity protection [56] and changing width per the spatial heterogeneity of the resistance surface. In general, for animals moving in a particular direction, prioritized areas with low ecological resistance and sudden ecological resistance changes can block progress. Further, the scope of corridors relates to the effective realization of species migration and information circulation among sources [57]. This paper uses the Ant algorithm and Kernel Density analysis to calculate the scope of ecological corridors, and the basic principle is that ants search for food resources by leaving pheromones to communicate with subsequent ants [58]. Following are the specific steps:

(1) Basic Data Preparation. Including ecological source points and the modified resistance surface. Ecological source points are central to the ecological source. They are the starting points for the ants to find the way. The resistance value of each pixel in the resistance surface represents the distance traveled by ants from the initial pixel to a given pixel, i.e., the higher resistance value of each pixel relates to the longer time spent by the ants spend in passing-through.

(2) Improved Ant Algorithm. In general, there may be certain problems when the ant colony algorithm is used to find the optimal path, such as: (1) If the movement of each ant is random, it will result in the situation that an ant falls into an infinite cycle in the local optimal solution. (2) Since the pheromones left by ants are discrete in space, it is difficult to determine the boundary of ecological corridor. For the first question, the angle between the direction of the current pixel ant and the next pixel is less than or equal to 90 degrees in the algorithm. The directional layer can reduce the blindness of ants in finding the way. For the second question, the results of ant colony algorithm and kernel density analysis are combined to identify the scope of ecological corridor. The specific steps are as follows:

Firstly, as per the minimum cumulative resistance surface calculated by the MCR model, the modified resistance surface determines the fundamental distribution of ecological corridors using the cost distance in ArcGIS10.2. Selection of some ecological corridor samples precedes establishing respective buffer zones of 300 m, 600 m, 900 m, 1200 m, 1500 m, 1800 m, 2100 m, 2400 m and 2700 m [59]. Repeated tests demonstrate, for the buffer zone larger than 1500 m, the pheromone density of the ants conforms to the spatial diffusion trend. Even though for less than 1500 m, the pheromone concentration range is too small to cover the buffer. Thus, the 1500 m buffer is the search area of the Ant algorithm. In this paper, the corridors data presents the specific situation of high resistance, which makes the ants unable to pass, and thereby, this paper improves the ordinary ant colony. The ants skip the points with higher resistance values and select points with lower resistance values for moving forward to reach the destination. Furthermore, using the rule of probability transition, every ant can update the pheromone globally. At every iteration, 200 ants start the path-finding algorithm, and after 50, 100 and 1000 iterations, it revealed that the pheromone concentration did not increase with the increase of iteration times. Thus, the paper selected fifty generations as the number of iterations for filtering the points higher to the global average pheromone concentration and obtain the globally optimal solution, as shown in Figure 3.

(3) Determination of ecological corridor scope and ecological restoration point. Transforming the pheromone left by the ants into vector points. The density surface construction was by the Kernel density analysis method. The quantiles divide results into nine grades. Concerning related papers [60], combined with the spatial characteristics of the density surface and the ecological resistance surface, revealed that the 7, 8 and 9 nuclear density surfaces have good spatial connectivity, and the change of the spatial form of the ecological corridor range, the rest of the nuclear density surfaces are broken. Therefore, the selected three highest levels for Kernel density results draw the scope of ecological corridors, regarded as chief ecological corridors, and the rest areas where ants find their way were potential ecological corridors. Furthermore, the resistance value mutation points or discontinuities separate the ecological corridor, and these points are regarded as the key points for restoring the ecological corridor, namely the ecological restoration points.

## 4. Results and Analysis

### 4.1. Spatial Distribution of Ecosystem Services and Analysis of Cold and Hot Spots

As shown in Figure 4, the overall gas regulation showed a decreasing trend from northeast to southwest to the central part, and the spatial distribution of high-value area was consistent with that of forest land. Similarly, the low-value is in the Liancheng Town built-up area, where the intensity of human activities is high, and the carbon sequestration capacity is relatively weak, as depicted in Figure 4a. Whole habitat maintenance also showed a decreasing trend from northeast to southwest to the central part with high values mainly distributed in Xiyang River, Tuoniang River, Qingshui River and the areas surrounding high vegetation coverage in Guangnan area of Nanli River Basin, as depicted in Figure 4b, or food supply. The high-value distribution is mainly in the surrounding areas of each township, where the terrain is flat and open, the water and soil resources and farming conditions are superior and the cultivated land has a wide distribution, including the food supply in Liancheng Town, Zhulin Town and Zhuanjiao Township is higher, as depicted in Figure 4c.

For water resources supply, as per the distribution of geomorphological characteristics, carbonate rocks distribution is wide in the western and southeastern Karst areas, owing to the specific soluble feature, widespread rock cracks, precipitation, surface runoff not being easy to store, the amount of infiltration being high and water resources supply being low. From the land type perspective, the fertile soil and strong rainfall infiltration capacity of forest land cause blocked surface runoff blocked and depletes water resources supply. The impervious surface area of construction land accounts for a large proportion, the precipitation infiltration capacity is weak, and the flood peak flow is large, making the water resources supply high, as depicted in Figure 4d. The widely distributed Karst landform of Guangnan County has a shallow soil layer, critical soil erosion, moderate water erosion and low overall soil conservation. The main distribution of high value is at Bambang Township and Zhuanjiao Township of Karst area in Northeast and Southwest Africa, and the soil conservation is lower in Babao Zhulin Karst area and Liancheng urban construction area, as depicted in Figure 4e. Combined evaluation results of the above ecosystem services select the top 20% patches of each ecosystem service as per the respective advantageous areas, and the spaces merge using the hot spot analysis tool in ArcGIS to obtain the ecosystem services of Guangnan County hot and cold areas, as depicted in Figure 4f. Overall, the concentration of hot spots is near the main rivers and non-Karst Forest lands in Guangnan, especially around the Guangnan area of Xiyang River in the east and the Forest land area in the northeast. Furthermore, the ecosystem service value around the big river and the stalagmite river in the southwest is also high in the hot spot area and the strongly distributed central part by human beings in the non-significant and cold spot regions, respectively.

### 4.2. Identification of Major Ecological Sources

Due to scattered and non-concentrated patches in the hot spot area, the distribution is sparse and the fragmentation degree is high, and according to Figure 5, the manual correction removes the patches with scattered distribution and area less than 6.26 km^2^ in the hot spot area as the ecological source. Lastly, twenty-three identified ecological sources have a total area of 1292.77 km^2^, accounting for 16.74% of the total regional area.

Overall, the main concentration of the ecological sources is in the Guangnan area of Xiyang River in the east, the forest land area in the northeast and the non-Karst area in the southwest and scattered in the northwest with high vegetation coverage, abundant animal and plant resources and relatively perfect ecosystem services, which has great significance to the ecological security of Guangnan County. It can provide a better living environment for local animals and plants. Among them, Bamei Town, Yangliujing Township and Nasa Town have the broadest distribution area, accounting for 36.69%, 18.38% and 11.72% of the total area of ecological sources, indicating that the ecological base of Bamei Town is the best, followed by Yangliujing Town, and Nasa Town is relatively weak. For the land cover types, the main ecological source patch is forest land, followed by cultivated land, accounting for 82.86% and 6.31% of the respective total area. The construction land and transportation land are not the ecological sources, and garden and grassland account for only a minor part. Concerning dominant ecosystem services, the areas providing one and two dominant ecosystem services account for 28.72% and 53.90% of the respective total ecological source area, and the mainland use type was forest land scattered in each source area. On the other hand, three and four services providing areas are in forest land and account for 16.52% and 0.86%, and mainly distributed in Yangliujing Township. Therefore, the ecosystem patches providing three or more dominant ecosystem services were fewer, indicating an apparent relationship between the trade-off and synergy of the ecosystem services.

### 4.3. Construction of Comprehensive Resistance Surface

Figure 6 depicts that the average base resistance value is 35.34, the maximum value is around 77.75 and the minimum value is around 12.16 with apparent spatial differentiation, which indicates an overall higher southwest to northeast. Karst area is widespread in Guangnan County with bad ecological environment and frequent natural disasters. Therefore, the constructed ecological sensitivity index modifies the foundation resistance surface. The rocky desertification in Guangnan County is grave and mainly concentrated in the southeastern part of the region, with Babao Town and Heiziguo Township as the most prominent. Guangnan County has a shallow soil layer and rich and uneven annual precipitation, which leads to soil erosion of different degrees in various Guangnan County regions. Among them, Bambang Township, Yangliujing Township and the northwest of Heiziguo Township are the most serious. The distribution of Karst collapse is consistent with the elevation distribution, with the northwest and southeast as the overall boundary and the northern risk is small, while the southern risk is high. Even though for Bamei Town, Liancheng Town and Yangliujing Township at the north of the boundary, the Karst collapse risks are relatively high, mainly due to the greater intensity of human activities causing ecological disturbances.

Due to the special natural background of the karst area, rocky desertification frequently occurs, a major natural disaster in the region and one of the important reasons hindering species migration. Therefore, the spatial distribution of the revised resistance value is consistent with that of rocky desertification. The modified resistance surface is along the line of “Zhe (Zetu Township)-Lian (Liancheng Town)-Yang (Yangliujing Township)-Ban (Bambang Township)” in space, with the characteristics of lower resistance value in the north and higher resistance value in the south. The most southern part of the Karst region has shallow soil territories, a high rock exposure rate, a single plant community and poor living conditions with mainly calcareous and scarce vegetation destroyed repeatedly. Thus, frequenting the regional soil erosion, rocky desertification and other natural disasters, and with the grim ecological restoration and governance situation, the ecology is highly fragile. Further, Guangnan County has a high poverty coefficient and agriculture-dominated production activities, so human activities have a higher interference with the ecological quality. Synthetically, the resistance value in the southern part of the line is high. The northern part is the main non-Karst area, with a better ecological base and good vegetation growth conditions, i.e., Bamei Town, the last Xanadu in China. Furthermore, the northwest of the region is sparsely populated and has low ecological environment pressure. Further, the implementation of ecological environment projects such as afforestation, returning farmland to forestry, and grassland also increases vegetation coverage. Thus, the resistance value in the north of the line is low.

### 4.4. Ecological Corridor and Ecological Restoration Point

Figure 7 shows the pheromone concentration left by ant colony algorithm, according to the spatial distribution characteristics of pheromones, they can be divided into the following four types:

(I) When the ecological corridors are at a high resistance value, the pheromone concentration left by the ants is low and scattered, as shown in Figure 7a. Similarly, when the ecological corridors are at a low resistance value, the pheromone concentration left by the ants is high and concentrated, as shown in Figure 7b.

(II) When the ecological corridors traverse from a low resistance value to a high resistance value, as shown in Figure 7c, the ants skip such points and choose the patches with a lower resistance value to advance, ensuring the movement of ants between two sources.

(III) When the ecological corridors are in the area where the local resistance value patches are relatively uniform and continuous, as shown in Figure 7d, before reaching the high resistance value, the ants tend to shift to the low resistance value, i.e., the pheromone points converge to the lower resistance patches near the high resistance value.

(IV) When the ecological corridors are in an area with high fragmentation of local resistance value patches and high heterogeneity, as shown in Figure 7e, ants mostly leave pheromones at the low resistance patches between two high resistance patches.

Thus, the pheromone spots left by ants mainly appeared in lower resistance patches, and the higher the pheromone concentration left by patches relates to the higher probability for the next ant choosing the patch.

An ecological security pattern based on the paradigm of “source corridor ecological restoration point” is constructed and shown in Figure 8. The ecological source areas are mainly distributed in the East and northeast of Guangnan County. These areas have high vegetation coverage, rich animal and plant resources and perfect ecosystem services, which are of great significance to the ecological security of Guangnan County and can provide a good living environment for the local animals and plants. There are 23 ecological source points, which are connected with the ecological corridor, which is of great significance to stabilize the regional ecosystem. There are twenty-three ecological corridors in the study area, in a ring distribution, with the total area of key ecological corridors of 804.95 km^2^, mainly distributed along with forest land and cultivated land passing through most of the nature reserves and main scenic spots and avoiding the towns and non-ecological land. The forest area accounts for 59.91%, followed by cultivated land, grassland and orchard, respectively, accounting for 19.47%, 7.54% and 5.64%.

The area of the potential ecological corridor is 621.2 km^2^, where the resistance value of the potential ecological corridor is higher and more fragile than the main ecological land. Thus, the strengthening of protection in potential ecological corridors is imperative to prevent the impact of human activities on the main ecological corridors. The corridors mainly located in the low resistance area have a high coincidence degree of main and potential ecological corridors, such as the ecological corridors in Bamei Town in the north of Guangnan County. When the corridors traverse from the low resistance value to the high resistance value area or mainly located in the high resistance value area, the width of the main ecological corridors is lower and surrounded by potential ecological corridors such as the ecological corridors in the northwest of Yangliujing Township and the north of Babao Town. In urban construction, the greening construction strengthening of ecological corridors can ensure the normal flow of species between sources and maintain ecological function continuity.

The thirty-eight identified ecological restoration points are mainly at the intersection of ecological corridors or prominent high resistance patches. Such areas largely fragment the connectivity of ecological corridors to a certain extent, making the morphological characteristics of ecological corridors quite different, hindering the transfer of material and energy between sources, and destroying the integrity and continuity of ecosystems and existing biological species. Therefore, leading to pertinent construction of tunnels, overpasses or culverts. Furthermore, other necessary biological and engineering measures improve ecological restoration sites, protect local biodiversity and improve the stability of the ecosystem.

## 5. Discussion

### 5.1. Discussion on Resistance Surface Construction in Karst Mountain Area

Yunnan Karst area is a vital part of the ecological shelter in southwest China, which plays a crucial role in stabilizing regional climate, protecting water resources and maintaining biodiversity [61]. However, due to the specific ecological conditions, coupled with the water and heat in the monsoon climate background and the high-intensity human activity interference, which emphasizes the regional ecologic fragility characteristics and direness of rocky desertification. This area has become the chief “depression” for China’s social and economic development and a key area for national poverty alleviation and ecological restoration [62]. Therefore, the construction of ecological resistance surface in the Karst area differs from others. It is necessary to construct a set of index systems in line with the regional characteristics following the specific background. Further, previous studies [63] only uniformly assign values to land-use types to determine the basic resistance surface, which inevitably conceals the difference in the impact of different land-use methods for resistance values under the same category and inaccurate reflections on the spatial heterogeneity of ecological resistance. They did not comprehensively consider the unique geographical characteristics of the study area; thus, it cannot accurately reflect the spatial heterogeneity of ecological resistance. Even though night light data [64] and impervious surface index [65] have been used to correct the resistance surface, it is difficult to reflect the typical regional ecological process, reflect the degree of disturbance in the fragile ecological environment area, let alone highlight the typical regional ecological problems. Therefore, as per the drawing lessons from former scholars on constructing the resistance surface [66], this paper sets peculiar ecological environment condition of the Karst area and an inherent mechanism for the rock desertification formation combining elevation, slope, land cover type, vegetation coverage, lithology and soil thickness of six thickness indexes for regional choice of Guangnan county to construct the basic resistance surface, and offers a reference for choosing the resistance factor of the Karst area.

Further, the poor soil condition of carbonate rock, shallow soil layer and high-quality soil resource shortage restrict the vegetation growth in Karst mountainous areas. Further, human beings have caused severe disturbances to Guangnan County, a poor agriculture-dominated mountainous county, which has aggravated the ecosystem fragility and has made restoration and management extremely difficult. The gradual degradation of ecology in some areas has induced many ecological disasters, including rocky desertification and soil erosion. Among them, the collapse of Karst is most significant. It illustrates that the ecological sensitivity of Karst areas has formed a few obstacles in the ecological source expansion. Therefore, the ecological sensitivity index modifies the basic resistance surface, which is more practical and conducive to the follow-up scientific construction of the ecological security pattern in Karst areas.

### 5.2. Ecological Corridor Scope Based on Ant Algorithm

As the chief carrier of material and energy flow between source patches, ecological corridors are vital channels for species migration, with social, cultural, ecological and other functions, and can provide several ecosystem services including flood storage, biodiversity protection, soil erosion prevention and so on. However, current studies are limited to the number and spatial distribution of ecological corridors and lack of quantitative methods to objectively determine the width threshold of ecological corridors, especially to study the possible impact of local background on species movement. In previous studies, the scope of the corridor was determined by referring to the movement radius of animals in the study area [67], but this method is only applicable to specific areas and species and is difficult to apply and adjust this method when different species and areas are involved [68]. Ant algorithm can simulate the best path of ants from the nest to the nearby food, i.e., the former ants will leave pheromones along the way to provide choices for the latter ants under the guidance of pheromones, ants form a positive feedback mechanism to improve the accuracy of the calculation results. In recent years, the Ant algorithm has gradually penetrated various fields, including the path planning of automatic vehicles [69] and the path planning of mobile robots [70]. So, the ant algorithm can be used as a choice to determine the scope of the corridors. The combination of Ant algorithm and Kernel density analysis can transform the discrete pheromone layer into a relatively smooth Kernel density layer and then determine the scope of the ecological corridor. In this paper, the improved general Ant algorithm is as follows:

(1) In the past, for studying the corridor range with the Ant algorithm, scholars usually set the initial point and endpoint of the ant’s movement and start the running, but obtaining the convergence result becomes difficult with the higher grid number. Peng et al. [61] set the maximum iterations at 60,000 generations, which stop with 100 ants reaching the endpoint or at no updating in the last 200 times. Under this method, few ants reach the endpoint. It also diffuses the pheromones left by ants making it hard to determine the best location of the corridor. Therefore, the Ant algorithm in this paper proposes the idea of dynamic programming in operational research [71], which divides the overall corridor optimization problem and converts it into individual and colony pathfinding in each corridor, i.e., dividing the larger grid into several acceptable small areas and running the Ant algorithm, so that the pheromone obtained would be relatively concentrated. Due to serially connected corridors, the independent optimal solution for every corridor can deduce the overall optimal solution. This method avoids the broader range inputs of ant path-finding space, which reduces the problem complexities, and improves algorithm efficiency and accuracy.

(2) In the general ant algorithm, ants move randomly to the surrounding eight adjacent pixels. However, the experimental course revealed that the phenomenon of ants spinning around in one place is as per the corrected resistance surface. Thus, this paper makes corresponding improvements, so ants have the “jumping” function and can move randomly in a nearby area instead of eight adjacent grids, which highly increases the probability of ants jumping out of local locations and reaching the endpoint. This method can provide solutions for the same problems encountered with running the ant colony algorithm.

### 5.3. Research Shortcomings and Prospect

This paper identifies the dominant patches of critical ecosystem services as the ecological source using hot spot analysis and constructs the resistance index system under the specific background of the Karst area, revised as per the ecological sensitivity index. By integrating the ant algorithm and Kernel density function, this paper delimits the scope of an ecological corridor and resolves the ecological corridor width issue. However, in Karst areas, besides the highly fragile ecology, intense human activities are also a significant factor hindering the ecological source expansions and the species source improvements. The paper only assesses the spatial distribution of regional resistance value from the perspective of ecological fragility, ignoring the impact of severe human interference in Karst areas. The former scholars used an impervious surface to improve the assessment accuracy and strengthen the ecological resistance value interpretation, but the impermeable surface starts from the land type and has limited interpretation ability of human actual disturbance intensity. To accurately quantify anthropogenic interference intensity, it is imperative to select better indicators such as the night light index [26]. Therefore, more attention to the ecological vulnerability and human disturbance intensity in the future can explore the spatial variation of ecological resistance value to provide strong support for the accurate ecological corridors and nodes identification. Next, using the ant algorithm to identify corridor width due to different corridor directions, aspect ratios and other parameters requires manually debug parameters. In the future, with the machine learning algorithm, the parameter value can reach the target points [72]. The majority of the current research on ecological security patterns uses the static perspective and identifies the various elements as per the status quo of the study area. However, regional development planning and ecological civilization construction are long-term, sustainable and dynamic processes, and the spatial distribution of related elements can significantly change with the regional economic development. Therefore, based on the current ecological security pattern, a long-term optimization scheme for ecological security patterns in the Karst area fulfill the regional ecological security requirement and strengthen the forward-looking ecological security pattern to ensure the stability of a regional ecosystem. Future research should focus on whether the existing ecological sources can meet the ecological security requirement with the sustainable regional economic development and the exiting damaged sources restoration to add important ecological land according to regional characteristics for scientifically increasing the radiation coverage of ecological sources. Simultaneously, increasing the ecological greening construction with important ecological corridors, enhancing the connectivity between sources, forming a pattern of urban development with ecological sources as the core and ecological corridors in series, stabilizing regional ecosystems and making them more ecologically livable.

## 6. Conclusions

For a typical Karst area in southeastern Yunnan, this paper focuses on Guangnan County to highlight ecological problems and determine the ecological source area per the chief ecosystem services. This paper starts from the regional ecologic environment conditions and uses the internal mechanism of rocky desertification to construct an index system for determining the basic resistance surface. Further, the ecological sensitivity index revises the comprehensive Ant algorithm and the nuclear density analysis, which determines the ecological corridor range and ecological restoration points and builds the county ecological security pattern to maintain the ecosystem stability in Guangnan County. The conclusions are as follows:

(1) The top 20% selected ecological patches from vital ecosystem services using hotspot analysis, identifies twenty-three ecological sources with a total area of 1292.77 km^2^, accounting for 16.74% of the total regional area. The main distribution is in the Guangnan watershed of Xiyang River in the east, forest land in the northeast and non-Karst area in the southwest, scattered in the northwest, where most wide distributions are in Bamei Town, Yangliujing Township and Nasa Town.

(2) The corrected resistance surface is with the line of “Zhe (Zhetu township)-Lian (Liancheng town)-Yang (Yangliujing township)-Ban (Banbang township)” in space with lower resistance value in the north due to concentration in the non-Karst area, superior the ecological base, improved vegetation growth condition, lower resistance value in the south due to the Karst area, a shallow soil layer and higher bare rock rate. Further, the plant community is single, the survival condition is poor, the ecological restoration and government situation are strict and the resistance value is high.

(3) Combining Ant algorithm and Kernel density analysis determines the key ecological corridor of 804.95 km^2^, the potential ecological corridor of 621.2 km^2^ and thirty-eight ecological restoration points. The construction of the ecological security pattern for “source-corridor-ecological restoration points” effectively improves the regional integrity and ecological sensitivity to provide strong support for sustainable development.

## Figures and Tables

**Figure 1 ijerph-18-06863-f001:**
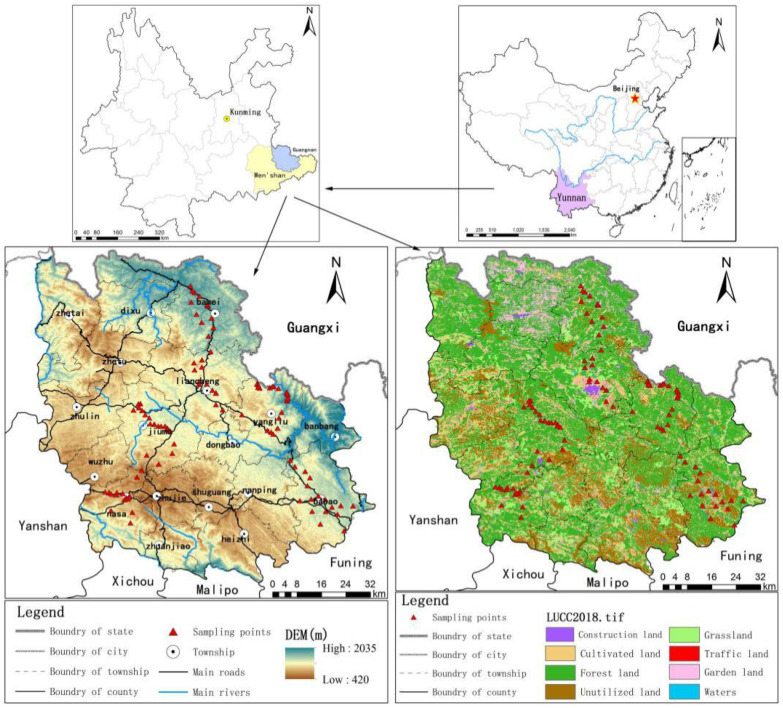
The geographical location of the study area.

**Figure 2 ijerph-18-06863-f002:**
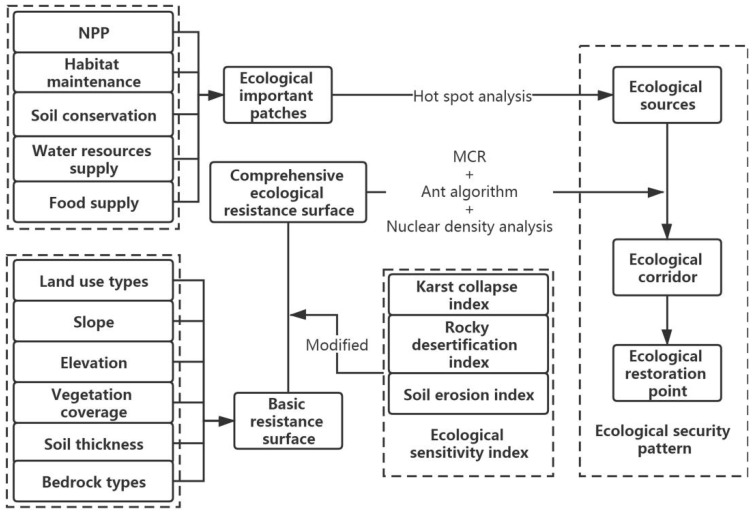
The methodology framework of the study.

**Figure 3 ijerph-18-06863-f003:**
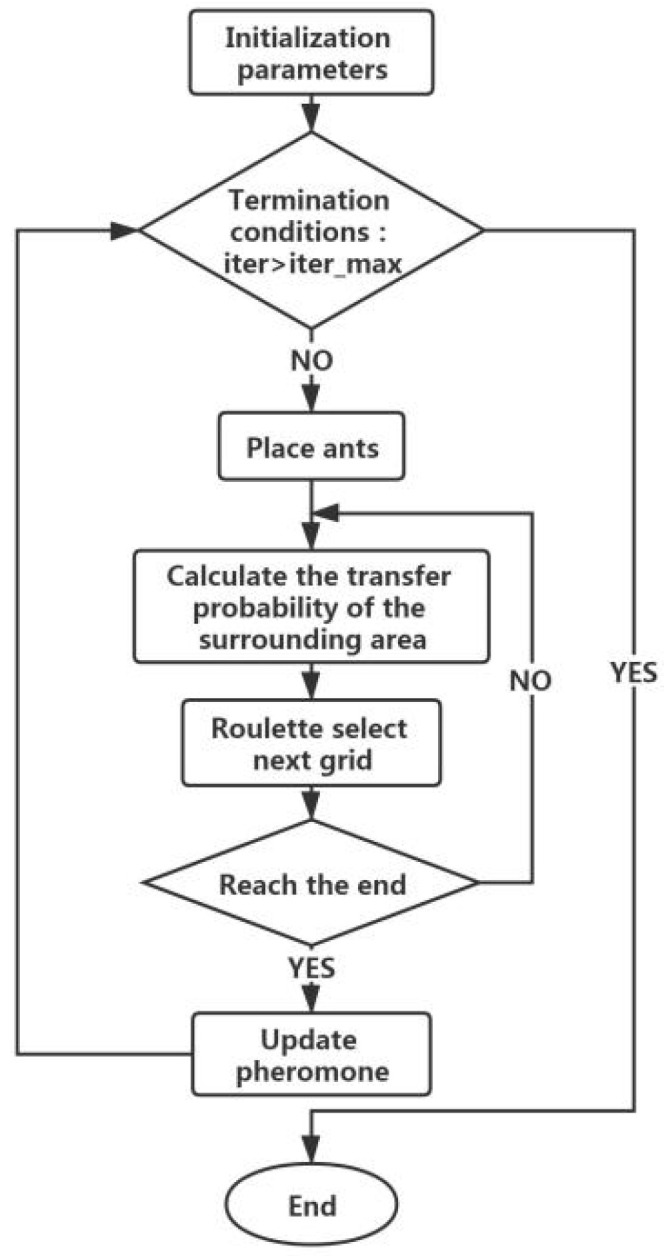
The technology route of ant colony algorithm.

**Figure 4 ijerph-18-06863-f004:**
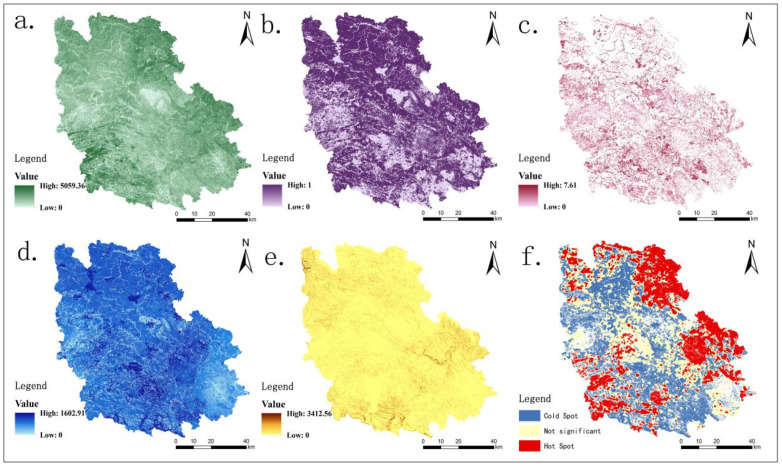
The spatial pattern of ecosystem services and the distribution of cold and hot areas. (**a**) NPP (gC·m^−2^·a^−1^); (**b**) Habitat maintenance; (**c**) Food supply (MkJ/km^2^); (**d**) Water resources supply (mm); (**e**) Soil conservation (t·hm^−2^·a^−1^); (**f**) Cold and hot areas.

**Figure 5 ijerph-18-06863-f005:**
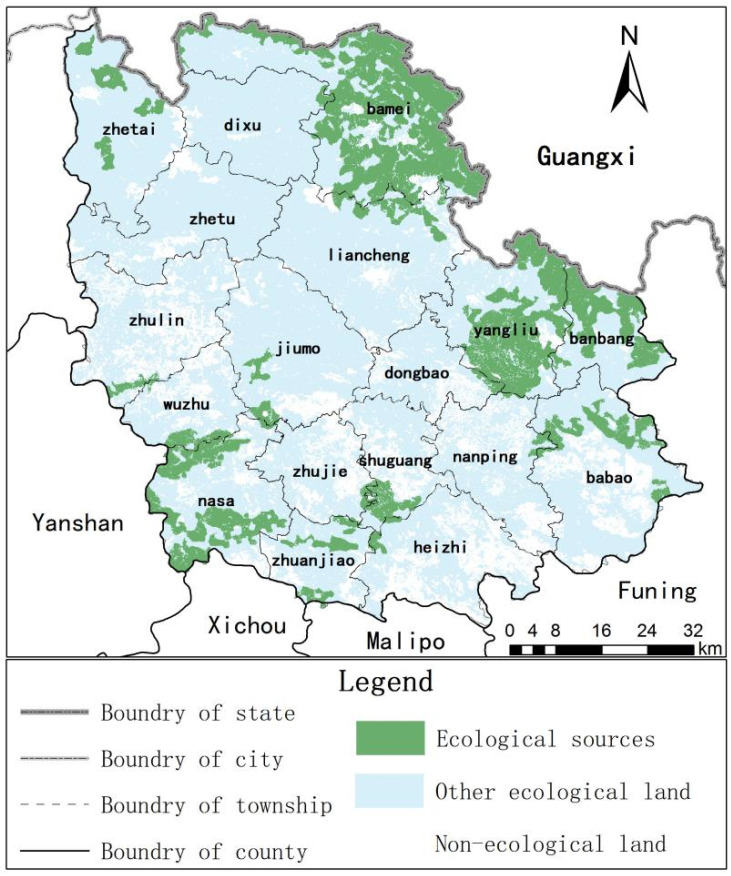
Spatial distribution of ecological sources.

**Figure 6 ijerph-18-06863-f006:**
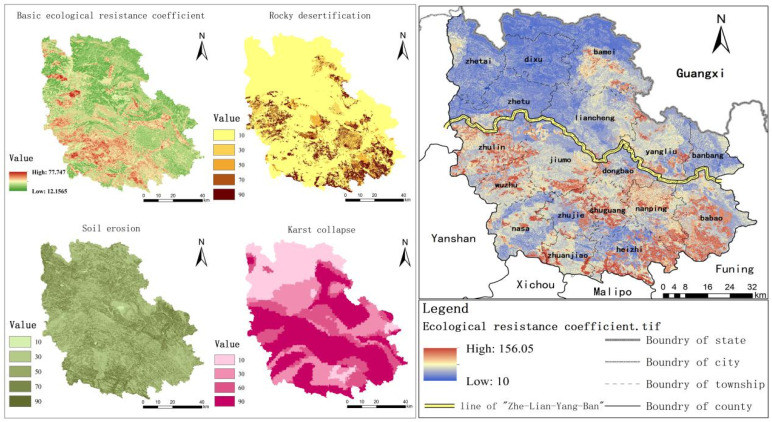
Construction of comprehensive resistance surface in Guangnan County.

**Figure 7 ijerph-18-06863-f007:**
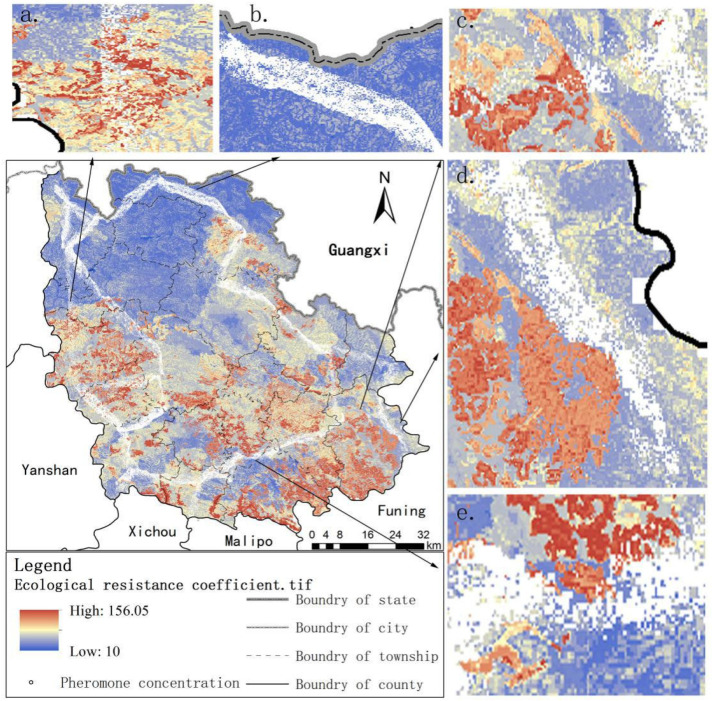
Spatial distribution of pheromone in Ant algorithm. (**a**–**e**) The ecological corridor is located in the same resistance value, and the pheromone concentration presents different characteristics in space.

**Figure 8 ijerph-18-06863-f008:**
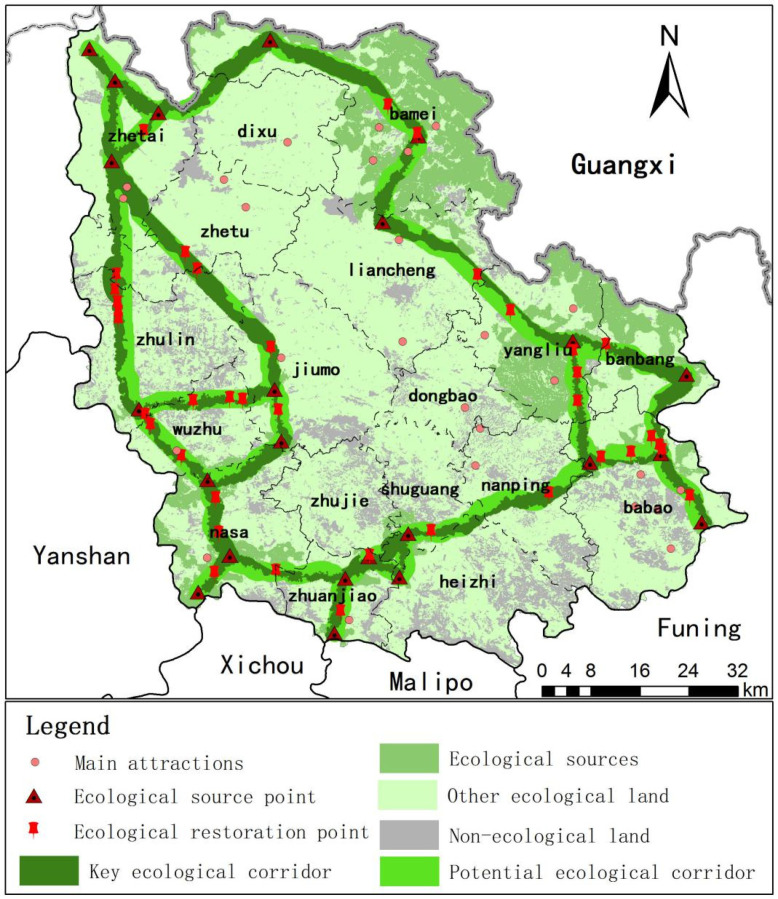
Ecological security pattern in Guangnan County.

**Table 1 ijerph-18-06863-t001:** Evaluation of ecosystem service.

Evaluation Types	Formula	Parameters
NPP	*NPP_t_* = *APAR_t_* × *ε_t_*	where *NPP_t_*, *APAR_t_* and *ε_t_* are the *NPP*, photo-synthetically active radiation absorbed by vegetation and light energy conversion rate of vegetation at spatial location x and time period *t*, respectively.
Habitat maintenance	$Q_{xj}=H_{j}\left(1-\frac{D_{xj}^{2}}{D_{xj}^{2}+k^{2}}\right)$	where *Qxj* and *Hj* are the habitat quality and habitat attribute of land use type *j*, respectively; *k* is a semi-saturated constant; *z* is a model default value; *Dxj* is habitat degradation.
Soil conservation	A=K×R×LS×(1−C×P)	where *A* represents the average annual soil conservation; *K* is soil erodibility factor; *P* is soil conservation measure factor; *R*, *LS* and *C* are the rainfall erosion factor, terrain factor and the cover-management factor, respectively.
Water resources supply	/	where W is the annual water resources supply, P, ET and Q are the annual precipitation, annual evapotranspiration and surface runoff, respectively.
Food supply	/	where ck is the average value of food supply energy per unit area in region k ; Pmk is the unit area yield of m crops in the region k ; Am is the energy of m crops

**Table 2 ijerph-18-06863-t002:** Resistance factor and weight in the study area.

Resistance Factors	Factor Weight	Resistance Classification	Basic Resistance Coefficient
Land cover type	0.1671	Forest land	5
		Grassland	10
		Garden Land	20
		Cultivated land	30
		Waters	50
		Unutilized land	70
		Traffic land	80
		Construction Land	100
Slope (°)	0.1198	0–15	10
		15–25	30
		25–35	50
		>35	80
Altitude (m)	0.2054	<800	10
		800–1300	30
		1300–1500	50
		>1500	80
Vegetation coverage (%)	0.1436	<35	10
		35–50	30
		50–65	50
		>65	80
Soil thickness	0.1496	<10	10
		10–30	40
		>30	70
Bedrock type	0.2145	Non-carbonate rock	10
		Limestone	30
		Interformation of limestone and dolomite	50
		Dolomite	70
		Carbonate rock with clastic rock	90

## Data Availability

The data presented in this study are available on request from the corresponding author.

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
