# Peer review of "Ecological Security Pattern Construction in Karst Area Based on Ant Algorithm"

_ijerph, 2021, doi:10.3390/ijerph18136863_

Round 1

Reviewer 1 Report

Dear Authors, the manuscript is potentially good but some points require in-depth revision.

I am curious with the ant algorith. Unfortunately, the manuscript is poor of information (only one paragraph 3.2.3). It is absolutely unsufficient for a paper grounded on this algorith. You have to provide an extensive literature review of that point, providing and discussing alternatives and giving the most relevant information on more algorithms. Why you have selected ants? Pros & cons in clear, please!

In general, I would see more literature review on the substantive problem. I don't like technical manuscripts with a strong calculation and no explanation about the reason for any philosophical choice. Please revise and enlarge the motivations of any choice adopted in this paper. The survey design needs also improvements.

Language usage requires a strong check. Shorter sentences in most cases are appreciated. Thank you.

Reviewer 2 Report

Interesting paper. The idea of authors is easy to follow. All parts of the paper are present and good elaborated. I have only minor comments:

- Ant Algorithm is not common to use i this type of study, so I suggest to describe the background of this statistics in methods

- Part 5 is “Discussion and Conclusion” and part 6 is “Conclusion”, I do not think that the 6th part is necessary

Reviewer 3 Report

introduction needs to conclude with what you plan to do and how it will affect the region you are targetting.

you need to be a bit clearer in how the applicaiton of your model provides targetted regions for protection. the selection of ecological corridoors is not so clear to the reader - the language is a little complex and makes understanding your work difficult. in figure 5 it looks like the resistance surface is dominated by the  rocky desertification classification.

Figure 6 needs to be clear what each of the sub maps is

figure 7 what are the ecological source points and restoration points. these are not very clear in the text.

Round 2

Reviewer 1 Report

I am satisfied of your revision. Thank you.